Methods

# In vivo and in vitro knockout system labelled using fluorescent protein via microhomology-mediated end joining

Shota Katayama[1,2] , Kota Sato[2,3], Toru Nakazawa[1,2,3,4]

**Gene knockout is important for understanding gene function and genetic disorders. The CRISPR/Cas9 system has great potential to achieve this purpose. However, we cannot distinguish visually whether a gene is knocked out and in how many cells it is knocked out among a population of cells. Here, we developed a new system that enables the labelling of knockout cells with fluorescent protein through microhomology-mediated end joining–based knock-in. Using a combination with recombinant adeno-associated virus, we delivered our system into the retina, where the expression of *Staphylococcus aureus* Cas9 was driven by a retina ganglion cell (RGC)–specific promoter, and knocked out *carnitine acetyltransferase* (*CAT*). We evaluated RGCs and revealed that *CAT* is required for RGC survival. Furthermore, we applied our system to *Keap1* and confirmed that *Keap1* is not expressed in fluorescently labelled cells. Our system provides a promising framework for cell type–specific genome editing and fluorescent labelling of gene knockout based on knock-in.**

## Introduction

Methods for gene knockout are required to understand the functions of genes and genetic disorders. The clustered, regularly interspaced, short palindromic repeat (CRISPR)/CRISPR-associated protein 9 (Cas9) system, which targets specific genomic loci and induces site-directed DNA breaks when combined with a single-guide RNA (sgRNA) that contains the complementary 20 nucleotides of the target sequence (Mojica et al, 2009; Garneau et al, 2010; Jinek et al, 2012; Wiedenheft et al, 2012; Cong et al, 2013; Hsu et al, 2013; Mali et al, 2013; Konermann et al, 2015; Ran et al, 2015), has been used for this purpose. However, it is not possible to visually determine whether gene knockout has occurred and how many knockout cells are present. Among the methods for gene delivery,

recombinant adeno-associated virus (rAAV) vector, which directly infects the retina after intravitreal injection, is effective in gene delivery to the retina (Pang et al, 2008). However, it was a concern that the entire size of the knockout system should be under 4.6 kbp for rAAV delivery (Colella et al, 2018), and retina ganglion cell (RGC)–specific gene knockout has not yet been achieved. A new system is, thus, needed that marks knockout cells with fluorescent protein and introduces gene knockout into RGCs specifically with rAAV.

To achieve this, we focused on microhomology-mediated end joining (MMEJ)–dependent integration of donor DNA using CRISPR/Cas9 (Nakade et al, 2014; Hisano et al, 2015). MMEJ requires an extremely short homologous sequence (5–25 bp) for DNA double strand break repair, resulting in precise integration into the targeted genomic loci (Katayama et al, 2016; Sakuma et al, 2016). MMEJ-mediated precise integration enables the development of a fluorescently labelled knockout system, in which a coding region between exon 2 and 5 is replaced with a fluorescent protein.

Glaucoma is characterized by a loss of RGCs (Quigley & Addicks, 1981; Jakobs et al, 2005; Kielczewski et al, 2005) and results in vision defects and blindness. The cause of RGC death is considered to have a genetic background (Shiga et al, 2017; Shiga et al, 2018), involving calpain activation (Ryu et al, 2012), oxidative stress (Himori et al, 2013), and ER stress (Yamamoto et al, 2014). Recently, metabolomic and histological analyses of mouse retina in an optic nerve crush model (RGC death like glaucoma) reported that L-acetylcarnitine levels were increased in the ganglion cell layer (GCL) (Sato et al, 2018). L-acetylcarnitine, which is synthesized from acetyl-CoA and carnitine by carnitine acetyltransferase (CAT) (Bieber, 1988; Liu et al, 2002), has neuroprotective and anti-oxidative effects (Jones et al, 2010). We speculated that L-acetylcarnitine has neuroprotective effects in RGCs, and *CAT* knockout promotes RGC death.

Kelch-like ECH-associated protein 1 (*Keap1*) is a transcription factor and a negative regulator of NFE2-related factor 2 (*Nrf2*) (Itoh et al, 1999; Kobayashi et al, 2004). *Nrf2* signaling is involved in counteracting oxidative stress in RGCs (Himori et al, 2013). *Keap1*

[1]Department of Advanced Ophthalmic Medicine, Tohoku University Graduate School of Medicine, Miyagi, Japan   [2]Department of Ophthalmology, Tohoku University Graduate School of Medicine, Miyagi, Japan   [3]Collaborative Program for Ophthalmic Drug Discovery, Tohoku University Graduate School of Medicine, Miyagi, Japan   [4]Department of Retinal Disease Control, Tohoku University Graduate School of Medicine, Miyagi, Japan

Correspondence: ntoru@oph.med.tohoku.ac.jp

knockdown up-regulates *Nrf2* signaling (Miyazaki et al, 2014) and induces a neuroprotective effect in RGCs.

In this study, we created a CRISPR-Cas9 platform that can be modulated using a cell type–specific promoter and can mark knockout cells with a fluorescent protein in vitro and in vivo. We show that this system can be used as a biomedical research tool.

## Results

### Construction of a knockout system labelled using iLOV protein in vitro

Considering the size limitations of the rAAV delivery system, we have to construct knockout systems less than ~4.6 kbp. Because the size of SaCas9 is 3.15 kbp (Ran et al, 2015) and Brn3b is an RGC-specific transcription factor (Sajgo et al, 2017; Zhang et al, 2017), we used the promoter sequence (−50 to −1, 50 bp) of *Brn3b* for SaCas9 expression (Fig 1A). The iLOV protein, which is a GFP with a small coding sequence of 336 bp (Chapman et al, 2008), was used as the knock-in protein (Fig 1A). We designed two vectors: *Control* and *Knockout* vectors (Fig 1A) for the *ROSA26* and *CAT* loci, respectively (Figs 1B and S7). The targeted sequences between *CAT* exon 2 and 5 are cut off by the sgRNA-Cas9 complex and have inserted the microhomology arm (MHA)–harboring DNA fragment, which contains a splicing acceptor (SA), for iLOV and synthetic polyA (pA), by MMEJ-dependent integration, thereby we can monitor knockout cells by measuring iLOV fluorescence (Fig 1B). In the *ROSA26* loci, the integrated locus is transcribed into pre-mRNAs (exon 2-intron-SA-P2A-iLOV-pA) and then matured to spliced mRNAs (exon 2-SA-P2A-iLOV-pA). The spliced mRNAs are translated into proteins (exon 2-SA-P2A-iLOV). In the *CAT* loci, the integrated locus is transcribed into mRNAs (exon 2-SA-iLOV-pA) and then translated into proteins (exon 2-SA-iLOV). The iLOV gene introduced in the targeted locus gets fluorescent for both Control and Knockout vectors.

To determine whether the designed sgRNAs edit the targeted regions, we constructed pX601-ROSA26 sgRNA, CAT sgRNA1, and sgRNA2 vectors and transfected them into Neuro2a cells. T7 endonuclease I (T7EI) assays revealed that ROSA26 sgRNA, CAT sgRNA1, and CAT sgRNA2 cut the genome at rates of 31%, 46%, and 20%, respectively (Fig 1C). To clarify whether our sgRNAs induce off-target mutations, we selected the two highest potential off-target sites of each gRNA, which were ranked using CRISPOR (http://crispor.tefor.net/) (Haeussler et al, 2016). We amplified the target sites by PCR and evaluated their sequences. There were no mutations in the potential off-target sites (Fig S1A).

In addition to *Brn3b* mini promoter–driven SaCas9 vectors, we generated *CMV* mini promoter (Ede et al, 2016)–driven SaCas9 vectors. *Brn3b* is highly expressed in Neuro2a cells and not expressed in NIH3T3 cells. We transfected these vectors into the cell lines and observed iLOV fluorescence in Neuro2a cells (Fig 1D). Although we observed iLOV fluorescence of cells transfected with *CMV* mini promoter–driven SaCas9 vectors, we observed no fluorescence of cells transfected with *Brn3b* mini promoter vectors in NIH3T3 cells (Fig S2A). We corrected that Control and Knockout vectors were precisely integrated into the targeted genomic locus in

Neuro2a cells (Figs 1E and S1B), not integrated in NIH3T3 cells (Fig S2B). Flow cytometry revealed that ~27% of Control cells were positive for iLOV fluorescence and about 22% of Knockout cells were positive (Fig S3, left panel). Combination with FACS, we can selectively collect the knockout cells with fluorescence. Therefore, iLOV fluorescence of about 20% are sufficient. We found the loss of *CAT* expression in iLOV-positive cells transfected with the Knockout vector by reverse transcriptase PCR (RT-PCR, Fig 1F) and quantitative PCR (qPCR, Fig 1G). Taken together, these data indicate that our system can knockout a gene with fluorescence depending on *Brn3b* expression.

### Knockout system labelled by iLOV protein in vivo

To determine whether our system can induce iLOV fluorescence in vivo, we applied this system to a retina GCL using AAV2. 5 wk after intravitreal injection, we observed iLOV fluorescence in GCL with both Control and Knockout vectors (Fig 2A), indicating that our system induces knockout in the targeted gene locus in vivo. We then calculated the knock-in efficiency of our system. We found that ~6% of cells in the Control GCL were positive for iLOV fluorescence and about 7% of the cells in the Knockout GCL were positive (Fig S4). The knock-in frequency of our system was ~6–7% in the GCL.

Microscopic observation using hematoxylin and eosin (HE) staining showed that the retina tissue contained abnormality in the GCL with *CAT* deficiency (Fig 2B). The GCL was immunolabelled for the RGC marker RBPMS, and the *CAT*-deficient GCL showed that the number of RGCs was remarkably decreased (Fig 2C). The cell density of RGCs in the *CAT*-deficient GCL was significantly reduced compared with in the Control GCL (Fig 2D). Taken together, these data demonstrated that *CAT* is required for RGC survival.

### Knockout of other endogenous genes by our system in vitro

To test whether our system allows the knockout of other genes, we generated a system targeting the mouse *Keap1* gene (Figs 3A and S7). To determine whether designed sgRNAs edit the targeted regions, we constructed pX601-Keap1 sgRNA1 and sgRNA2 vectors, and transfected them into Neuro2a cells. T7E1 assays revealed that Keap1 sgRNA1 and sgRNA2 cut the genome at 22% and 17%, respectively (Fig 3B). To clarify whether Keap1 sgRNAs induce off-target mutations, we selected the two highest potential off-target sites of each gRNA, which were ranked using CRISPOR (http://crispor.tefor.net/). We amplified the target sites by PCR and evaluated their sequences. There were no mutations in the potential off-target sites (Fig S5A). We transfected Control and Knockout vectors into Neuro2a cells and observed iLOV fluorescence (Fig 3C). Flow cytometry revealed that ~26% of Control cells were positive for iLOV fluorescence and about 16% of Knockout cells were positive (Fig S3, right panel). Control and Knockout vectors were precisely integrated into the targeted genomic locus (Figs 3D and S5B). We confirmed that knockout of *Keap1* expression in the Knockout vector transfected and iLOV+ cells by RT-PCR (Fig 3E) and qPCR (Fig 3F). Together, these data support the applicability of our system to other endogenous genes.

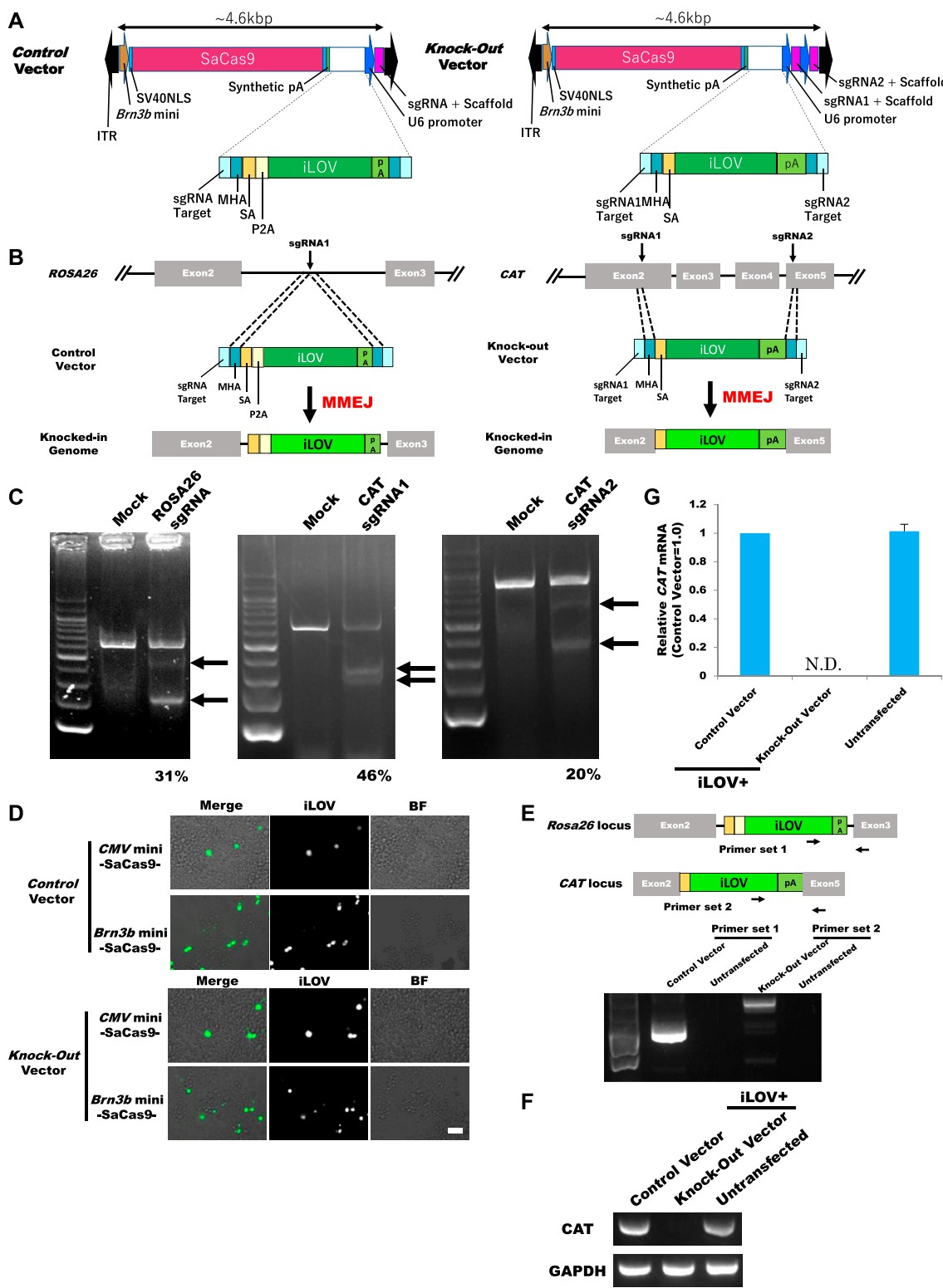

**Figure 1. Knockout of the *CAT* gene using the Brn3b promoter–driven knockout system plasmid.**
**(A)** Schematic model of the Control and Knockout vectors. **(B)** Scheme of the fluorescently labelled knockout system. **(C)** T7E1 assay for each sgRNAs. Arrows: size of cleaved fragments (also indicated below). %: quantified editing efficiency. **(D)** Fluorescence microscopic observation at 3 d after transfection. Scale bar: 50 $\mu$m. **(E)** PCR evaluation of the knockout system using the primers shown. **(F)** RT-PCR analysis of *CAT* transcription. **(G)** qPCR analysis of *CAT* transcription. Error bars indicate SD (*n* = 3). *Brn3b* mini, *brn3b* minimal promoter; *CMV* mini, *cmv* minimal promoter; ITR, inverted terminal repeat; N.D., not detected; P2A, porcine teschovirus-1 2A; SV40NLS, sv40 NLS.

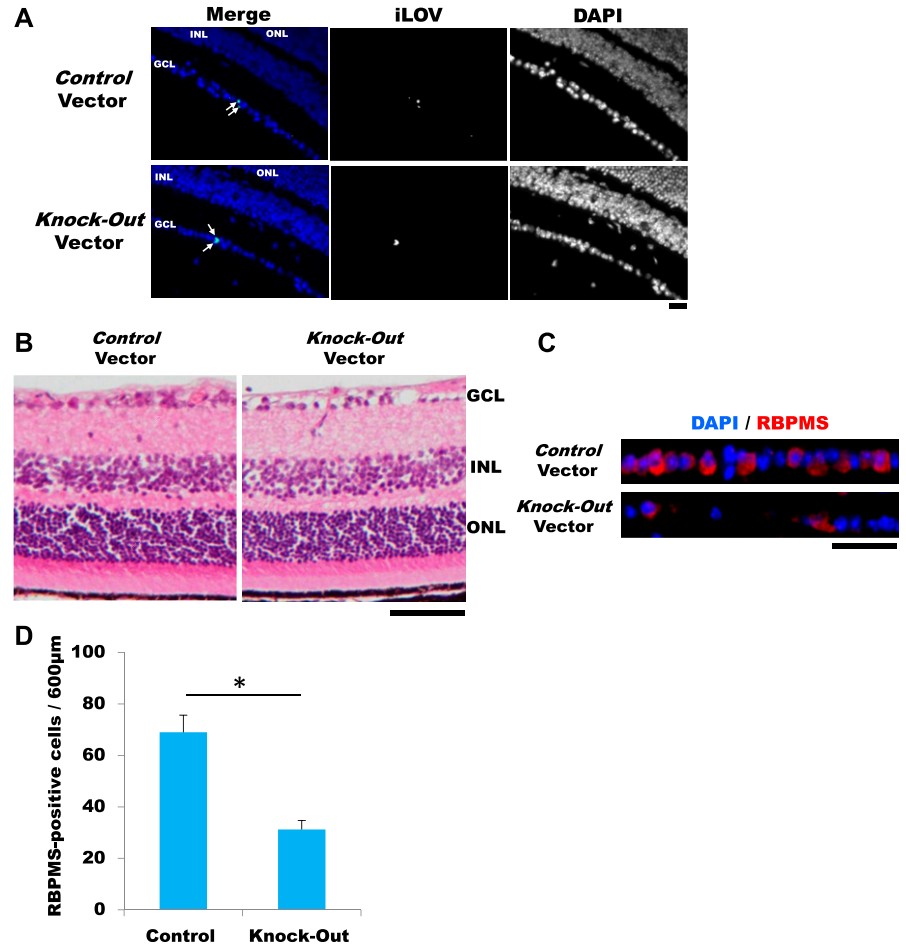

**Figure 2.** *CAT* deficiency in the GCL using a AAV2/*Brn3b* promoter-driven knockout system.
**(A)** Fluorescence microscopic observation at 5 wk after intravitreal injection. White arrowheads indicate iLOV-positive cells. Nuclei were stained with DAPI (blue). Scale bar: 50 μm. **(B)** HE staining. Scale bar: 50 μm. **(C)** Immunostaining for the RGC marker RBPMS (Alexa Fluor 594, red). Nuclei were stained with DAPI (blue). Scale bar: 50 μm. **(D)** The number of RBPMS+ RGC in a 600-μm region of the retina from the inferior side of the optic nerve head. Error bars indicate SD (n = 3), *P < 0.05 (two-tailed *t* test). GCL, ganglion cell layer; INL, inner nuclear layer; ONL, outer nuclear layer.

## Discussion

In this study, we successfully designed a fluorescent knockout system that can be modulated by a cell type–specific transcription factor. In this design, SaCas9 is driven by a *Brn3b* mini promoter, and iLOV is integrated into the targeted genomic loci through MMEJ.

An MMEJ-dependent strategy has been used for various applications such as disease modeling in human-induced pluripotent stem cells (Kim et al, 2018) or the generation of knock-in mice harboring an exogenous gene (Aida et al, 2016). MMEJ is a precise and efficient knock-in method. Although what we detected are error rates (*ROSA26*; 3 clones of 15 clones, *CAT*; 2 clones of 15 clones, *Keap1*; and 4 clones of 15 clones, Figs S1B and S5B), our system is not influenced by an error rate at the 3′ junction because the 3′ junction is behind the termination of RNA polymerase II transcription. Although the insertion of a fluorescent cassette takes place at only one allele and not biallelic, the targeted gene knockout takes place because both or one of the two kinds of CRISPR cut in the targeted locus (Figs S1B and S5B). Therefore, our system can correctly knockout the targeted gene through MMEJ.

iLOV is not a commonly used fluorescent protein. Potential toxicity lined to *iLOV* expression was considered. We performed Annexin V/PI double staining assay and revealed that early apoptotic (Annexin V+/PI−) cells are not increased by *iLOV* expression (Fig S6). Thus, *iLOV* expression is not toxic.

Our approach has potential limitations. The target gene needs to be expressed at sufficient levels for detection by FACS of the iLOV transgene; gRNA efficiency and sequence features of the short homology arms may be critical to promote efficient repair by MMEJ. In addition, the proportion of clones with only one knock-in event and efficient knockout of the remaining allele may be difficult to predict.

Several studies have shown that L-acetylcarnitine has a neuroprotective effect in patients who have experienced hypoxic-ischemic brain injury (Zanelli et al, 2005) and against oxygen-glucose deprivation in neural stem cells (Bak et al, 2016). On the other hand, L-acetylcarnitine deficiency induced serious deleterious effects on the central nervous system (Virmani & Binienda, 2004). Our results also showed that reduced L-acetylcarnitine by the loss of *CAT* induced RGC death. These findings suggested that CAT-mediated L-acetylcarnitine production is required for the maintenance of homeostasis in nerve cells.

Our results presented herein indicate that our knockout system could be applied to elucidating gene function and genetic disorders in vitro and in vivo, especially in the field of ophthalmology.

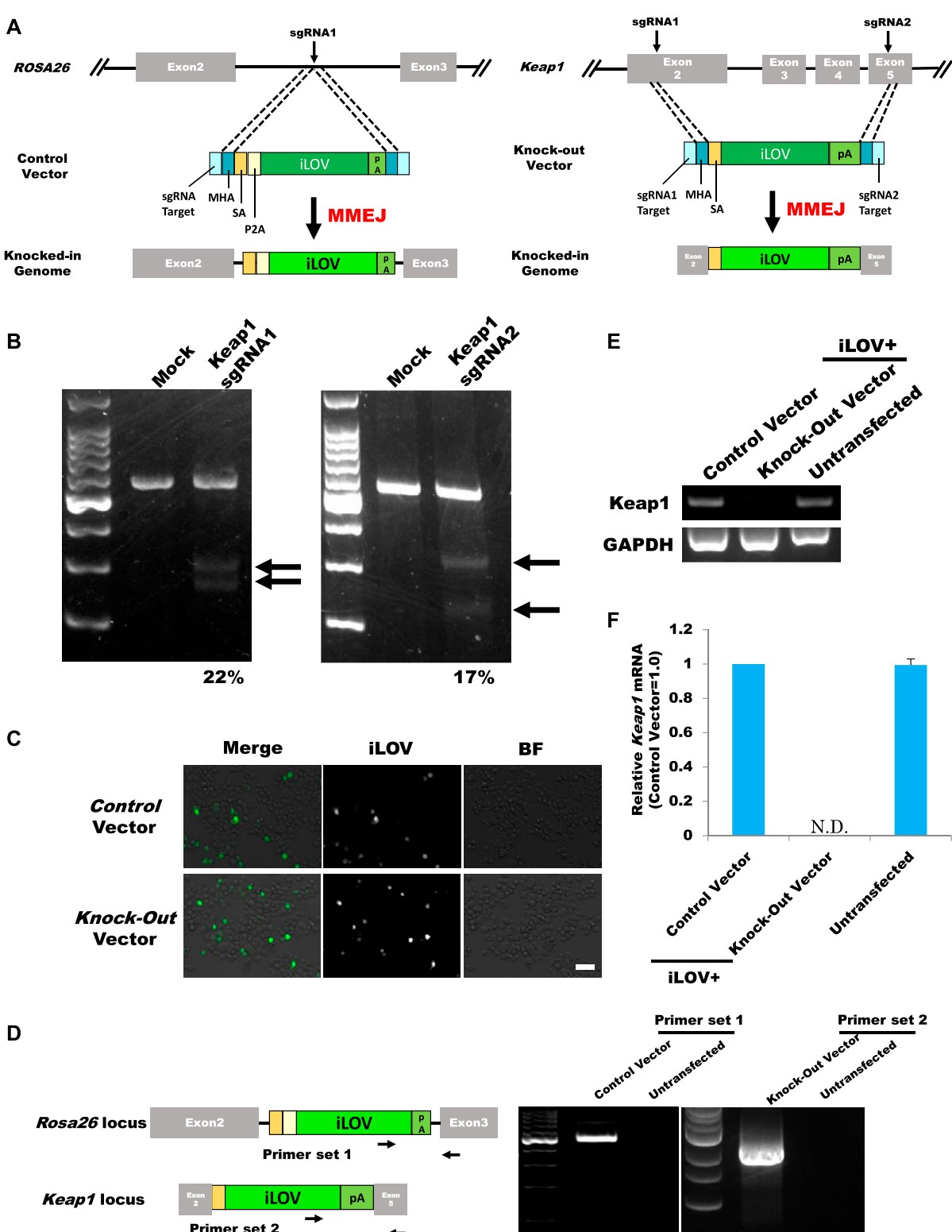

**Figure 3.   Knockout of the *Keap1* gene by using the *Brn3b* promoter–driven knockout system plasmid.**
**(A)** Scheme of the fluorescently labelled knockout system. **(B)** T7E1 assay for *Keap1* sgRNAs. Arrows: size of cleaved fragments (also indicated below). %: quantified editing efficiency. **(C)** Fluorescence microscopic observation at 3 d after transfection. Scale bar: 50 μm. **(D)** PCR evaluation of the knockout system using the primers shown. **(E)** RT-PCR analysis of *Keap1* transcription. **(F)** qPCR analysis of *CAT* transcription. Error bars indicate SD (*n* = 3). N.D., not detected.

# Materials and Methods

## Vector construction

The CRISPR/Cas9 (*Streptococcus aureus* Cas9, SaCas9) plasmid was constructed using the single vector AAV-Cas9 system (Ran et al, 2015). Oligonucleotides for sgRNA templates were synthesized, annealed, and inserted into the corresponding vectors. pX601 vectors for the mouse ROSA26, CAT (exon 2 and 5), and Keap1 (exon 2 and 5) genes, termed pX601-ROSA26 sgRNA, CAT sgRNA1, CAT sgRNA2, Keap1 sgRNA1, and Keap1 sgRNA2, were constructed. Oligonucleotide sequences are shown in Table S1.

To construct knock-in vectors, we modified pX601-AAV-CMV::SaCas9-NLS-P2A -GFP-NRPpA-U6::BsaI-sgRNA vector. First, we changed the CMV promoter to a CMV mini or *Brn3b* mini promoter and SV40 NLS by PCR using KOD ONE (Toyobo). The resulting fragments were assembled using Gibson Cloning HiFi DNA Master Mix (NEB) (pX601-AAV- CMVmini or *Brn3b*mini::SV40 NLS-SaCas9-NLS-P2A-GFP-NRPpA-U6::BsaI- sgRNA). Second, we inserted synthetic pA and SA by PCR and Gibson cloning (pX601-AAV- CMVmini or *Brn3b*mini::SV40 NLS-SaCas9-NLS-synthetic pA-SA-P2A-GFP-NRPpA- U6::BsaI-sgRNA). Third, we amplified iLOV sequences from pEiLOV-N1 (Addgene) and changed GFP to iLOV by PCR and Gibson cloning (pX601-AAV-CMVmini or *Brn3b*mini::SV40 NLS-SaCas9 -NLS-synthetic pA-SA-P2A-iLOV-NRPpA-U6::BsaI-sgRNA). Fourth, we changed NRPpA to synthetic pA by PCR and blunt end cloning (Toyobo) (pX601-AAV-CMVmini or *Brn3b*mini::SV40 NLS-SaCas9-NLS-synthetic pA-SA-P2A-iLOV-synthetic pA-U6::BsaI-sgRNA). Control vector: Fragment 1 (pX601-AAV-CMVmini or *Brn3b*mini::SV40 NLS-SaCas9-NLS- Synthetic pA), Fragment 2 (sgRNA target-MHA-SA-P2A-iLOV-synthetic pA-MHA-sgRNA target), and Fragment 3 (U6::ROSA26 sgRNA) were assembled with Gibson cloning. Knockout vector: Fragment 1 (pX601-AAV-CMVmini or *Brn3b*mini::SV40 NLS-SaCas9-NLS- synthetic pA), Fragment 2 (sgRNA1 target-MHA-SA-iLOV-Synthetic pA-MHA-sgRNA2 target), and Fragment 3 (U6::CAT (Keap1) sgRNA1 and U6::CAT (Keap1) sgRNA2) were assembled with Gibson cloning. All plasmids were verified using BigDye Terminator Kit version 3.1 (Applied Biosystems) and an ABI sequencer model 3130xl (Applied Biosystems). Primer sequences are shown in Table S2. The annotated DNA sequences of all-in-one cassettes are shown in Fig S7.

## Cell culture and DNA transfection

Neuro2a and NIH3T3 cells were cultured at 37°C in 5% $CO_2$ in Dulbecco's modified Eagle's medium (Nacalai Tesque) supplemented with 10% fetal bovine serum (HyClone), 100 U/ml of penicillin, and 100 µg/ml of streptomycin (Gibco). Lipofectamine 2000 (Life Technologies) and Opti-MEM (Life Technologies) were used for transfection according to the manufacturers' instructions. The plasmid concentrations, cell numbers, and plates used were as follows: 300 ng for pX601-ROSA26 sgRNA, CAT sgRNA1, CAT sgRNA2, Keap1 sgRNA1, and Keap1 sgRNA2 vectors, 100 ng for the pcDNA3-EGFP vector, and 300 ng for the pBrn3b (pCMV)-ROSA26, CAT and Keap1 vectors into $0.5 \times 10^5$ cells using a 24-well plate. pX601 vectors and pcDNA3-EGFP vector were cotransfected.

## RNA isolation and reverse transcription

Total RNA was purified from 5-d-old Neuro2a cells (iLOV-sorted) after transfection with Qiazol reagent (QIAGEN). One microgram of total RNA was used for the reverse transcription reaction with a transcriptor first-strand cDNA synthesis kit (Roche) in according to the manufacturer's instructions. RT-PCR helped display the representative gel image of the three independent replicates. qPCR was performed with Power SYBR Green Master mix (Applied Bioscience) and analyzed with the 7300 real-time PCR system (Applied Biosystems). Primer sequences are shown in Table S3.

## T7E1 assay

Genomic DNA was extracted from 48-h-old Neuro2a cells (EGFP-sorted) after transfection. The target site was amplified using PCR with the appropriate primer set (Table S4). The PCR amplicon was purified using a Nucleospin kit (MN). Then, 200 ng of each amplicon is diluted to 10 µl with 1× Oligo Annealing buffer. The amplicon is denatured and rehybridized in a thermal cycler programmed to incubate at 95°C for 10 min followed by 1 min each at 85°C, 75°C, 65°C, 55°C, 45°C, 35°C, and 25°C. Then, 3 µl DDW, 1.5 µl 10× NEB2 buffer and 0.5 µl 10 U/µl T7E1 (NEB) are added, and the reactions are incubated at 37°C for 30 min. The resulting products were analyzed by electrophoresis in 3% agarose gel and were visualized with Gel Red. The intensity of the bands of the PCR amplicon and cleavage products was measured using ImageJ (NIH). The efficiency was calculated using the following formula: % gene modification = 100 × $(1 - [1 - \text{fraction cleaved}]^{1/2})$.

## Genomic PCR and off-target analyses

Genomic DNA was extracted from Neuro2a and NIH3T3 cells using a QIAamp DNA Mini Kit (QIAGEN) according to the manufacturer's instructions and then used for PCR. Primer sequences are listed in Tables S4 and S5.

We used the CRISPOR (http://crispor.tefor.net/) to identify off-target candidate sites for the sgRNAs. DNA sequencing of PCR-amplified candidate sites was performed as mentioned above. Primers are shown in Table S5.

## Flow cytometry

3 d after transfection, the cells were harvested with 0.25% trypsin–EDTA and washed with culture medium. After washing with culture medium, the cells were suspended with FACS buffer (PBS and 2 mM EDTA). Flow cytometry analysis was performed using a FACS Aria II machine (BD) using a dual-wavelength analysis (488 nm solid-state laser and 638 nm semiconductor laser).

## Single-cell cloning

3 d after transfection, Neuro2a cells were dissociated and sorted into a single-cell/well (96-well plates) using a FACS Aria II machine (BD). 2–3 wk after sorting, living clones were picked up and genomic DNA extracted for PCR.

### Animals

4-wk-old male C57BL/6J mice were purchased from SLC and were maintained at Tohoku University Graduate School of Medicine with a 12-h light/dark cycle. The animal experiments in this study were performed in accordance with the Association for Research in Vision and Ophthalmology (ARVO) statement for the Use of Animals in Ophthalmic and Vision Research and were approved by the institutional animal care and use committee of Tohoku University, following the Guidelines for Animals in Research.

### AAV vectors and injection

Recombinant AAV (rAAV)2/2. Control and Knockout vectors were generated and purified in accordance with the method described previously (Fujita et al, 2015). Briefly, the Control or Knockout plasmid, AAV2 serotype–specific packaging plasmid, and helper plasmid, in a ratio of 1:1:3, were mixed with polyethyleneimine (PEI; Polysciences Inc.) and incubated for 10 min to form complexes. The transfection complexes were added to HEK293T cells and left for 72 h. The cells were harvested and lysed by freeze and thaw (3×) in PBS. AAV2 was bound to an AVB Sepharose column (GE Healthcare) and eluted with 50 mM glycine, pH 2.7, into 1M Tris, pH 8.8. AAV2 were washed with PBS and concentrated to a volume 100–150 $\mu$l using Vivaspin 4 concentrators. The viral vector was reconstituted at $1.0 \times 10^9$ genome copy/ml and used for intravitreal injection (2.0 $\mu$l/ injection) into mice aged 5 wk old.

### Hematoxylin and eosin (H&E) staining

H&E staining of mouse retina cryosections was performed as described previously with minor modifications (Ikuta et al, 2017). Briefly, hematoxylin staining was performed with hematoxylin solution (Type M; Muto Pure Chemicals) and eosin staining was performed with 0.3% eosin alcohol solution (Muto Pure Chemicals).

### Immunohistochemistry

Staining with anti-RBPMS antibodies was performed as described previously with slightly modifications (Sato et al, 2018). Briefly, rabbit anti-RBPMS (#194213; dilution 1:200; Abcam) was used as a primary antibody for a 1-h reaction at room temperature and then donkey antirabbit IgG Alexa Fluor 594 (molecular probe #A21207, 1: 500) was used as the secondary antibody.

### Annexin V/PI double staining

Annexin V and PI double staining was carried out with Annexin V Apoptosis Detection Kit (BD) according to the manufacturer's instructions. The samples were analyzed using a FACS Aria II machine (BD).

### Statistical analyses

Data were analyzed using a two-tailed $t$ test, with significant differences defined as $P < 0.05$.

## Supplementary Information

## Acknowledgements

This work was supported by JSPS KAKENHI (to 17H04349 T Nakazawa and 18H06248 S Katayama) and by the grant from Tohoku University Graduate School of Medicine (to S Katayama).

### Author Contributions

S Katayama: conceptualization, data curation, formal analysis, funding acquisition, investigation, visualization, methodology, project administration, and writing—original draft, review, and editing.
K Sato: data curation and methodology.
T Nakazawa: conceptualization, supervision, funding acquisition, and writing—original draft, review, and editing.

### Conflict of Interest Statement

The authors declare that they have no conflict of interest.

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
