## [Reviewer comments · Life Science Alliance]

Life Science Alliance

Knock-out system labelled using fluorescent protein via microhomology-mediated end joining

Shota Katayama, Kota Sato, and Toru Nakazawa

DOI: <https://doi.org/10.26508/lsa.201900528>

Corresponding author(s): Toru Nakazawa, Tohoku University

Review Timeline:

Submission Date:	2019-08-20
Editorial Decision:	2019-09-20
Revision Received:	2019-11-19
Editorial Decision:	2019-12-04
Revision Received:	2019-12-12
Accepted:	2019-12-12

Scientific Editor: Andrea Leibfried

Transaction Report:

September 20, 2019

Re: Life Science Alliance manuscript #LSA-2019-00528-T

Toru Nakazawa
Department of Ophthalmology, Tohoku University Graduate School of Medicine
1-1, Seiryō-cho
Sendai 980-8574
Japan

Dear Dr. Nakazawa,

Thank you for submitting your manuscript entitled "In vivo and in vitro knock-out system labelled using fluorescent protein via microhomology-mediated end joining" to Life Science Alliance. The manuscript was assessed by expert reviewers, whose comments are appended to this letter.

As you will see, the reviewers appreciate your method and provide constructive input on how to further strengthen your study and validate your results by including further controls. We would thus like to invite you to submit a revised version of your manuscript to us, addressing the comments made by the reviewers.

Thank you for this interesting contribution to Life Science Alliance. We are looking forward to receiving your revised manuscript.

Sincerely,

Andrea Leibfried, PhD

Executive Editor
Life Science Alliance
Meyerhofstr. 1
69117 Heidelberg, Germany
t +49 6221 8891 502
e a.leibfried@life-science-alliance.org
www.life-science-alliance.org

B. MANUSCRIPT ORGANIZATION AND FORMATTING:

Reviewer #1 (Comments to the Authors (Required)):

The authors show a straight-forward method properly backup by experimental data, which appears sufficient for this new, rapid-communication, journal.

Textual:

- it is not possible to visually determine whether gene knock-out has occurred and how many

knock-out cells are present because the knock-out cells are not labelled with a fluorescent protein.

- o □ Rewrite; because fluorescent labelling is just one means of detecting a correct KO.
- ...20 nucleotides of the target sequence and the protospacer-adjacent motif.
- o □ Rewrite; now it reads like the sgRNA contains a PAM.
- ...entire size of the knock-out system should be under 4.6 kbp for recombinant adeno-associated virus (rAAV) delivery.
- o □ Rewrite; There are multiple means to introduce the Cas9/sgRNA complex. One way is via rAAV-mediated delivery. Please insert one sentence describing this and why this one is chosen.
- Keap1 knockdown upregulates Nrf2 signaling.
- o □ How physiologically amenable is a KO instead of a knock-down described in literature? Is this gene a critical gene or not? If no significant detrimental effect is expected then include this in the text.

Furthermore,

The authors show that the percentage of iLOV fluorescence is 22% of Knock-out positive cells. 100% would make it a hugely interesting tool but this is not attainable thus far. It would be appreciated if the authors include some reasoning why this yield of 22% is sufficient and what are the reasons for this percentage (MMEJ efficiency?).

Reviewer #2 (Comments to the Authors (Required)):

Katayama et al. developed a knockout system based on microhomology-mediated endo joining. Its main point is that the knockout system enables the labeling of knockout cell with a fluorescent protein, known as iLOV. The authors have shown that the knockout system actually works in vitro and in vivo as exemplified for two different gene targets, CAT and Keap1. The reviewer evaluates this work is interesting. A comments is raised in the following:

Comment-1

The explanation about how SaCas9 introduces iLOV gene into the targeted locus is well described. However, the explanation about why the iLOV gene introduced in the targeted locus get fluorescent is unclear and not easy to understand. Because this is the main point of the presented technology, the authors are required to give more detailed explanation about how the iLOV gene introduced in the targeted locus get fluorescent for both Control and Knock-Out vectors.

Reviewer #3 (Comments to the Authors (Required)):

Katayama et al constructed all-in-one vectors for gene knock-out with the CRISPR-Cas9 system. The vectors include expression cassettes for S Aureus Cas9 and guide RNA as well as a donor sequence for microhomology-directed repair of the double strand break introduced by Cas9. The donor sequence designed here contains a cassette for expression of the iLOV fluorescent protein in order to facilitate identification of knock-in cells by fluorescence.

Insertion of an expression cassette for a marker protein is a common approach in model organisms (such as *D. melanogaster* or *C. elegans*) because it greatly facilitates identification and propagation of mutant animals. When trying to do gene knock-out in cultured cells or, in vivo, in somatic cells,

however, there is a big caveat to this approach: cells with marker expression can have insertion at either a single or both alleles and marker expression does not necessarily correspond with gene inactivation. When insertion takes place at only one allele, the second may be wild-type or mutated by end-joining, which might inactivate the target gene if the reading frame is disrupted. Indeed a conceptually similar approach to Katayama et al was reported by Wassef et al doi.org/10.1016/j.jymeth.2017.05.003, using antibiotic resistance as a marker, and only 10 to 30% of resistant clones carried biallelic targeting. The authors should also quote [doi: 10.1080/21655979.2017.1313645](https://doi.org/10.1080/21655979.2017.1313645) which previously proposed to achieve gene knock-out by MMEJ-mediated knock-in of a fluorescent expression cassette.

Consequently, in the approach proposed here, iLOV expression is not expected to directly correspond with gene inactivation and additional experiments are needed to better document the mechanisms and efficiency of gene knock-out:

- in cultured cells, target gene expression in iLOV-positive cells should be quantified by RT-qPCR in order to better examine the level of gene inactivation at the mRNA level (Figure 1f and 3e).
- more clones iLOV-positive clones should be genotyped in order to better quantify the level of gene inactivation at the DNA level. In particular, the data should show compound heterozygous mutant and homozygous mutant clones.
- both in cultured cells and in vivo, a larger number of target genes should be tested.

iLOV is not a commonly used fluorescent protein. Potential toxicity linked to iLOV expression should be examined and discussed.

Minor comments

1) What is the time course of iLOV expression in cells? AAV DNA is known to persist for a long time after transduction. Are gene knock-out efficiencies increased by waiting for longer than 3 days? Conversely, since AAV is a single stranded DNA vector, is it necessary to wait for a minimum amount of time?

2) The methods section should be more detailed. The iLOV fluorescent protein is little used and annotated DNA sequences of all-in-one cassettes should be provided with iLOV sequence, sgRNA target sites, micro-homologies, etc.... Information on rAAV production is also needed. The reference provided for rAAV production (Fujita et al, 2015) only quotes another paper from the lab detailing AAV production of another serotype (Nishiguchi et al, 2015).

3) The text needs to be checked. For example,

p2 Delete "because CRISPR-Cas9 knock-out cells are not fluorescently labeled".

p3 the sgRNA contains 20 nucleotides complementary to the target sequence but does not contain the protospacer adjacent motif, as stated by the authors.

p3 Delete "because the knock-out cells are not labeled with a fluorescent protein".

...

4) Figures legends are not sufficiently detailed. For example, legends of schematics in Figures 1a, 1b ... should include abbreviations used. Figure 1 a shows the schematic of AAV cassettes but experiments in Figure 1 were not done with AAV but plasmid DNA...

Responses to the Reviewer 1:

We thank this reviewer's comments. We have made changes in the revised manuscript.

The authors show a straight-forward method properly backup by experimental data, which appears sufficient for this new, rapid-communication, journal.

Textual:

- it is not possible to visually determine whether gene knock-out has occurred and how many knock-out cells are present because the knock-out cells are not labelled with a fluorescent protein.

o ∅ Rewrite; because fluorescent labelling is just one means of detecting a correct KO.

We appreciate to this reviewer's comment. As requested by the reviewer 3, we have removed the phrase: "because the knock-out cells are not labelled with a fluorescent protein".

- ...20 nucleotides of the target sequence and the protospacer-adjacent motif.

o □ Rewrite; now it reads like the sgRNA contains a PAM.

Thank you very much. We have removed the phrase: "and the protospacer-adjacent motif".

- ...entire size of the knock-out system should be under 4.6 kbp for recombinant adeno-associated virus (rAAV) delivery.

o ∅ Rewrite; There are multiple means to introduce the Cas9/sgRNA complex. One way is via rAAV-mediated delivery. Please insert one sentence describing this and why this one is chosen.

We agree to this reviewer's comment. We have added the sentence: "Among the methods for gene delivery, recombinant adeno-associated virus (rAAV) vector, which directly infects the retina after intravitreal injection, is effective gene delivery to the retina."

- Keap1 knockdown upregulates Nrf2 signaling.

o ∅ How physiologically amenable is a KO instead of a knock-down described in literature? Is this gene a critical gene or not? If no significant detrimental effect is expected then include this in the text.

We appreciate to this reviewer's comment. *Keap1* knock-out mice dies postnatally because of oesophageal hyperkeratosis (Ref.1). *Keap1* knock-out accelerates the nuclear localization of Nrf2 and its nuclear localization affects the expression levels of several squamous epithelial genes (Ref.1). Thus, *Keap1* is a critical gene in mice. Interestingly, *Keap1* knockdown constitutively accelerates the nuclear localization of Nrf2 without lethality (Ref.2). In this case, the nuclear accumulation of Nrf2 is weaker than that of the *Keap1*-deficient mice (Ref.2). Therefore, depending on the tissue/cell-type, knock-out may have a physiological adverse effect, but knock-out approach is necessary to elucidate the function of the targeted gene.

Reference

1. Wakabayashi et al., *Nat. Genet.* 35, 238–245 (2003).
2. Taguchi et al., *Mol. Cell. Biol.* 30, 3016–3026 (2010).

Furthermore,

The authors show that the percentage of iLOV fluorescence is 22% of Knock-out positive cells. 100% would make it a hugely interesting tool but this is not attainable thus far. It would be appreciated if the authors include some reasoning why this yield of 22% is sufficient and what are the reasons for this percentage (MMEJ efficiency?).

We thank this reviewer for his/her nice suggestion. We have added the sentence: "Combination with fluorescence-activated cell sorter (FACS), we can selectively collect the knock-out cells with fluorescence. Therefore, iLOV fluorescence of about 20% is sufficient." The reason for this percentage is MMEJ efficiency.

Responses to the Reviewer 2:

We thank this reviewer's comments. We have made changes in the revised manuscript.

Katayama et al. developed a knockout system based on microhomology-mediated endo joining. Its main point is that the knockout system enables the labeling of knockout cell with a fluorescent protein, known as iLOV. The authors have shown that the knockout system actually works in vitro and in vivo as exemplified for two different gene targets, CAT and Keap1. The reviewer evaluates this work is interesting. A comments is raised in the following:

Comment-1

The explanation about how SaCas9 introduces iLOV gene into the targeted locus is well described. However, the explanation about why the iLOV gene introduced in the targeted locus get fluorescent is unclear and not easy to understand. Because this is the main point of the presented technology, the authors are required to give more detailed explanation about how the iLOV gene introduced in the targeted locus get fluorescent for both Control and Knock-Out vectors.

Thank you very much for his/her nice comment. We have added the paragraph: “In the *ROSA26* loci, the integrated locus is transcribed into pre-mRNAs (exon 2-intron-SA-P2A-iLOV-pA) and then matured to spliced mRNAs (exon 2-SA-P2A-iLOV-pA). The spliced mRNAs are translated into proteins (exon 2-SA-P2A-iLOV). In the *CAT* loci, the integrated locus is transcribed into mRNAs (exon2-SA-iLOV-pA) and then translated into proteins (exon2-SA-iLOV). The iLOV gene introduced in the targeted locus gets fluorescent for both Control and Knock-Out vectors.”

Responses to the Reviewer 3:

We thank this reviewer’s comments. We have performed additional experiments and have made changes in the revised manuscript and figures.

Katayama et al constructed all-in-one vectors for gene knock-out with the CRISPR-Cas9 system. The vectors include expression cassettes for S Aureus Cas9 and guide RNA as well as a donor sequence for microhomology-directed repair of the double strand break introduced by Cas9. The donor sequence designed here contains a cassette for expression of the iLOV fluorescent protein in order to facilitate identification of knock-in cells by fluorescence.

Insertion of an expression cassette for a marker protein is a common approach in model organisms (such as D. melanogaster or C. elegans) because it greatly facilitates identification and propagation of mutant animals. When trying to do gene knock-out in cultured cells or, in vivo, in somatic cells, however, there is a big caveat to this approach: cells with marker expression can have insertion at either a single or both alleles and marker expression does not necessarily correspond with gene inactivation. When insertion takes place at only one allele, the second may be wild-type or mutated by end-joining, which might inactivate the target gene if the reading frame is disrupted.

Indeed a conceptually similar approach to Katayama et al was reported by Wassef et al doi.org/10.1016/j.ymeth.2017.05.003, using antibiotic resistance as a marker, and only 10 to 30% of resistant clones carried biallelic targeting. The authors should also quote doi: 10.1080/21655979.2017.1313645 which previously proposed to achieve gene knock-out by MMEJ-mediated knock-in of a fluorescent expression cassette.

We appreciate to this reviewer's comments. We have designed two CRISPR plasmids for the targeted locus and replaced the coding sequences with a knock-in cassette through MMEJ-mediated knock-in. Wassef et al described that they designed single CRISPR plasmid for the targeted locus and used HDR-mediated knock-in not MMEJ-mediated knock-in. Double CRISPR are more efficient knock-out than single CRISPR. MMEJ is more efficient knock-in than HDR. If the insertion of a fluorescent cassette takes place at only one allele not biallelic, the targeted gene knock-out takes place because both or one of the two kinds of CRISPR cut the targeted locus (Figs S1B and S5B).

Nakamae et al described that single CRISPR cut the targeted loci and a *CMV*-driven fluorescent cassette is integrated through MMEJ. In this approach, even if the insertion of a fluorescent cassette takes place imprecisely (frame shift or untargeted loci), *CMV*-driven fluorescence is observed. Our system does not have fluorescence when the unprecise integration (frameshift or untargeted loci) takes place. In terms of precise fluorescent knock-out, our system is a better method than single CRISPR/*CMV*-driven fluorescence.

Consequently, in the approach proposed here, iLOV expression is not expected to directly correspond with gene inactivation and additional experiments are needed to better document the mechanisms and efficiency of gene knock-out:

- in cultured cells, target gene expression in iLOV-positive cells should be quantified by RT-qPCR in order to better examine the level of gene inactivation at the mRNA level (Figure 1f and 3e).

We agreed to this reviewer's comment. We have performed RT-qPCR and revealed that iLOV-positive cells don't have the targeted mRNA expression as shown in Figs 1G and 3F. We have added the sentence in Main manuscript on page 6 and Methods on page 2.

- more clones iLOV-positive clones should be genotyped in order to better quantify the level of gene inactivation at the DNA level. In particular, the data should show compound heterozygous mutant and homozygous mutant clones.

Thank you very much for his/her nice suggestion. We have generated 15 iLOV-positive clones in each all-in-one vector. We have revised Figs S1B and S5B, and inserted the sentence in Main manuscript on page 8~9.

- both in cultured cells and in vivo, a larger number of target genes should be tested.

Single-cell cloning and qPCR analysis revealed that iLOV fluorescence is directly corresponding with gene inactivation. If the insertion of a fluorescent cassette takes place at only one allele not biallelic, the targeted gene knock-out takes place because both or one of the two kinds of CRISPR cut the targeted locus. We confirmed this fact on the targeted genes (*CAT* and *Keap1*). We think two target genes are enough to confirm whether iLOV fluorescence is corresponding with gene inactivation.

iLOV is not a commonly used fluorescent protein. Potential toxicity linked to iLOV expression should be examined and discussed.

We agree to this reviewer's comment. We have performed Annexin V assay and revealed that *iLOV* expression is not toxic in mouse cells. We have added Fig S6, and inserted the sentences in Main manuscript on page 9.

Minor comments

1) What is the time course of iLOV expression in cells? AAV DNA is known to persist for a long time after transduction. Are gene knock-out efficiencies increased by waiting for longer than 3 days? Conversely, since AAV is a single stranded DNA vector, is it necessary to wait for a minimum amount of time?

Since stable transgene expression is observed 2-4 weeks after AAV2 injection in mice (Sarra et al., *Vision Research* 2002), we have to wait for 2-4 weeks after intravitreal injection.

In vitro experiments, all-in-one plasmids are expressed 24-48 hours after transfection. iLOV expression is observed 48-72 hours after transfection. Gene knock-out efficiencies are not increased by waiting for longer than 3 days (data not shown).

2) The methods section should be more detailed. The iLOV fluorescent protein is little

used and annotated DNA sequences of all-in-one cassettes should be provided with iLOV sequence, sgRNA target sites, micro-homologies, etc.... Information on rAAV production is also needed. The reference provided for rAAV production (Fujita et al, 2015) only quotes another paper from the lab detailing AAV production of another serotype (Nishiguchi et al, 2015).

We have added the annotated DNA sequences of all-in-one cassettes (Fig S7). Further, we have described more detail in AAV production on Methods file.

3) The text needs to be checked. For example,

p2 Delete "because CRISPR-Cas9 knock-out cells are not fluorescently labeled".

We have removed this phrase.

p3 the sgRNA contains 20 nucleotides complementary to the target sequence but does not contain the protospacer adjacent motif, as stated by the authors.

As requested by the reviewer 1 and reviewer 3, we have revised.

p3 Delete "because the knock-out cells are not labeled with a fluorescent protein".

We have removed this sentence.

4) Figures legends are not sufficiently detailed. For example, legends of schematics in Figures 1a, 1b ... should include abbreviations used. Figure 1 a shows the schematic of AAV cassettes but experiments in Figure 1 were not done with AAV but plasmid DNA...

Thank you very much for this comment. We have added abbreviations and transduction methods in Figure legends.

We wish to express our thanks to the reviewers for providing these highly constructive comments. With these additional modifications, we hope that the manuscript is now acceptable for publication. Thank you very much.

December 4, 2019

RE: Life Science Alliance Manuscript #LSA-2019-00528-TR

Toru Nakazawa
Department of Ophthalmology, Tohoku University Graduate School of Medicine
1-1, Seiryō-cho
Sendai 980-8574
Japan

Dear Dr. Nakazawa,

Thank you for submitting your revised manuscript entitled "Knock-out system labelled using fluorescent protein via microhomology-mediated end joining". As you will see, reviewer #3 appreciates the introduced changes and we would thus be happy to publish your paper in Life Science Alliance pending final revisions necessary to alter the discussion and to adhere to our formatting guidelines:

- Please address the remaining reviewer concern
- Please link your profile in our submission system to your ORCID iD, you should have received an email with instructions on how to do so
- Please be more explicit in your conflict of interest statement
- Please note that figures cannot span multiple pages, please fix Figure 1
- Please upload all figure files, including suppl figure files as separate files; the legends to all figures (also to suppl figures) should go into the main manuscript file; please split Figure S3 and S4 for upload of individual files
- Please mention in the figure legend to figure 2 the statistical test used
- Please make sure that all components of the figures are also described in the legends (eg, for Fig1C and 3B the %)
- Please add a callout to figure S7 and add a legend to this figure in the main ms file
- Please incorporate the Methods section into the main manuscript file; please be more explicit in the methods section on number of replicates performed in the experiments (eg, for RT-PCRs)
- Please use our reference style - our typesetters can also work with EMBO journal style, should this be easier for your reference manager

A. FINAL FILES:

B. MANUSCRIPT ORGANIZATION AND FORMATTING:

Sincerely,

Reviewer #3 (Comments to the Authors (Required)):

The authors have satisfactorily addressed the issues raised in my review and I now recommend publication. The manuscript provides an elegant approach for isolating KO cells for a target gene of interest. The discussion may be improved to include potential limitations. For example, the target gene needs to be expressed at sufficient levels for detection by FACS of the iLOV transgene, guide RNA efficiency and sequence features of the short homology arms may be critical to promote efficient repair by MMEJ. In addition, the proportion of clones with only one KI event and efficient KO of the remaining allele may be difficult to predict.

Responses to the Reviewer 3:

We thank this reviewer's comments. We have made changes in the revised manuscript.

The authors have satisfactorily addressed the issues raised in my review and I now recommend publication. The manuscript provides an elegant approach for isolating KO cells for a target gene of interest. The discussion may be improved to include potential limitations. For example, the target gene needs to be expressed at sufficient levels for detection by FACS of the iLOV transgene, guide RNA efficiency and sequence features of the short homology arms may be critical to promote efficient repair by MMEJ. In addition, the proportion of clones with only one KI event and efficient KO of the remaining allele may be difficult to predict.

We appreciate to this reviewer's comment. We have inserted potential limitations in discussion section.

We wish to express our thanks to the reviewers for providing these highly constructive comments. With these additional modifications, we hope that the manuscript is now acceptable for publication. Thank you very much.

December 12, 2019

RE: Life Science Alliance Manuscript #LSA-2019-00528-TRR

Prof. Toru Nakazawa
Tohoku University
Department of Ophthalmology, Tohoku University Graduate School of Medicine
1-1, Seiryō-cho
Sendai 980-8574
Japan

Dear Dr. Nakazawa,

Thank you for submitting your Methods entitled "Knock-out system labelled using fluorescent protein via microhomology-mediated end joining". It is a pleasure to let you know that your manuscript is now accepted for publication in Life Science Alliance. Congratulations on this interesting work.

DISTRIBUTION OF MATERIALS:

Again, congratulations on a very nice paper. I hope you found the review process to be constructive and are pleased with how the manuscript was handled editorially. We look forward to future exciting submissions from your lab.

Sincerely,
